# OpenReview forum: "The Attacker Moves Second: Stronger Adaptive Attacks Bypass Defenses Against LLM Jailbreaks and Prompt Injections"
_ICLR.cc/2026/Conference — Submitted to ICLR 2026_

### Official Review · Reviewer_kWyh · 2025-10-24

**Soundness:** 3
**Presentation:** 3
**Contribution:** 3
**Rating:** 6
**Confidence:** 3

**Summary:**

The Authors critique the standard evaluation practices for LLM jailbreak and prompt injection defenses, arguing that testing against static attack datasets provides a false and misleading sense of security. The authors posit that defenses must instead be evaluated against strong, adaptive attackers who are aware of the defense's design (mirroring Kerckhoffs's Principle from cryptography). This paper introduces a general, four-step adaptive attack framework (Propose, Score, Select, Update, or "PSSU") and instantiate it using four powerful attack families: gradient-based, reinforcement learning, search-based, and large-scale human red-teaming. By applying this rigorous methodology, the authors systematically "break" 12 recent and diverse LLM defenses, achieving over 90% attack success rates on systems that originally reported near-zero vulnerability. The work concludes that the field must adopt this more adversarial and adaptive evaluation standard to make credible claims of robustness.

**Strengths:**

1. The paper's primary strength is its comprehensive and rigorous experimental takedown of 12 recent LLM defenses.
2. The "PSSU" (Propose, Score, Select, Update) loop  is a clear and effective generalization that unifies disparate attack methods (gradient, RL, search, human) into a single, understandable conceptual framework.
3. The unique and impressive strength is the execution of a large-scale ($20,000 prize pool, 500+ participants) human red-teaming competition.

**Weaknesses:**

1. The paper's automated attacks, particularly the search-based method, are extremely powerful, relying on a SOTA LLM (Gemini-2.5 Pro) as the "LmMutator" and another LLM as a "critic". While this is a fair choice under their "large compute" threat model, it somewhat obscures the "algorithmic" contribution of the PSSU framework itself. It is difficult to disentangle whether the high ASR is due to the novel search/RL framework or the emergent creative-reasoning capabilities of the SOTA models powering the attack.

2. The paper correctly identifies reward hacking and unreliable auto-raters as a key weakness in automated evaluation ( Appendix C.1 ). However, the search-based attack's "Scorer" component uses a "separate critic LLM" to generate numerical scores from qualitative feedback. This appears to re-introduce the very problem the paper warns about. The authors do not sufficiently discuss how they validated this critic LLM or ensured it wasn't providing misleading or "hackable" scores, which could in turn lead the LmMutator down trivial, non-generalizable paths.

**Questions:**

Question-1: Your search-based attack's 'Scorer' relies on a 'separate critic LLM' to convert textual feedback into a numerical score. How did you validate this critic LLM to ensure it wasn't susceptible to its own form of reward hacking (i.e., the LmMutator learning to "trick" the critic) and that its scores meaningfully guided the search toward genuinely effective attacks?

Question-2: how much of the automated attack's success hinges on using a SOTA model (Gemini-2.5 Pro) as the LmMutator? Have you tested whether the PSSU framework's effectiveness persists if a weaker, open-source model is used as the mutator? This would help clarify the specific contribution of your framework versus the raw capability of the powerful model you used to drive it.

---

> ### Author Response · Authors · 2025-11-25
>
> Thank you for the review and suggestions.
>
>
> Q1.  We agree that any optimization process can, in principle, induce reward hacking. However, in our setup the LLM critic is used only during the optimization phase, not as the final evaluator of attack success. Final success is determined using programmatically verifiable criteria, not LLM judgments. Therefore, even if the mutator were to “hack” the critic during search, it would not count as a successful attack unless it actually satisfies the ground-truth success condition.  Moreover we did manual checks for all of the results to ensure everything was correct.
>
> Q2. We view the primary contribution of our work as advocating for adaptive evaluation frameworks rather than relying on static benchmarks. While stronger models like Gemini-2.5 Pro certainly help produce higher-quality mutations, this is not the fundamental reason our attacks succeed. The core issue is that the evaluated defenses rely on fixed, non-adaptive attacks. One of the key messages of our work is that no single attack—ours included—should be used as the sole robustness metric, and defense designers should instead evaluate with tailored adaptive attacks using the strongest available methods. Moreover, we see that attacks like GCG (that have been known) when used in the correct setting can still break the attacks we evaluated (Appendix A.1).
>
> —-
>
> We have some preliminary results on the search attack against Gemini 2.5 Pro as the target model. We use the same genetic algorithm and the same setup but swap in different models as the mutator and the critic LLMs.
>
> | Mutator & Critic LLM  | ASR (%) |
> |:----------------------|:-------:|
> | Gemini 2.5 Pro        | 100     |
> | Gemini 2.5 Flash      | 91      |
> | Gemini 2.5 Flash Lite | 89      |
> | Gemma 3 27B           | 76      |
> | Qwen 3 235B A22B      | 100     |
> | Qwen 3 32B            | 91      |
> | Qwen 3 4B Thinking    | 39      |
>
> We observe that larger and “better” models (as measured by other capabilities) are also a better attacker LLM. That said, much smaller open-weight models like Qwen 3 32B and Gemma 3 27B still make up a strong adaptive attack.

---

> > ### Comment · Reviewer_kWyh · 2025-11-28
> > **Final Comments on Rebuttal**
> >
> > Thank you for answering all my concerns. I'll not increase my score at this point, but I'll push for acceptance during the discussion phase with ACs

---

### Official Review · Reviewer_XpmW · 2025-10-26

**Soundness:** 2
**Presentation:** 2
**Contribution:** 2
**Rating:** 2
**Confidence:** 3

**Summary:**

This paper challenges current evaluation practices for LLM defenses, arguing that robustness claims based on static attack sets or weak optimization methods are unreliable. The authors propose evaluating defenses against adaptive adversaries that tailor their strategies to the defense design and use scalable optimization methods such as gradient descent, RL, random search, and human-guided exploration. Applying this framework, they successfully bypass 12 prominent LLM defenses, each originally reporting near-zero attack success, now with success rates exceeding 90%. Their findings highlight that robustness requires testing against adaptive, defense-aware attackers, calling for stronger and more realistic evaluation protocols.

**Strengths:**

1. The paper addresses a highly relevant and pressing issue in LLM safety, evaluating the true robustness of jailbreaking and prompt-injection defenses.
2. The authors conduct an extensive and systematic analysis of 12 diverse defense mechanisms, providing a thorough assessment.

**Weaknesses:**

1. The proposed attack framework requires more clarification for different threat models. For example, under white-box access, what exactly is the attacker optimizing? Are they generating a suffix, a full prompt template, or token-level perturbations, and which optimization strategies are available or suitable in each case? Conversely, under black-box access, what are the precise inputs and outputs for each attack family (gradient-based, RL, search, human red-teaming), and how do those strategies interact with limited-query, rate-limited environments? Detailing these distinctions would improve the paper’s quality.
2. Lack of methodological details. Although the paper introduces a generalized PSSU framework, the description remains largely conceptual. Critical implementation details, such as how prompts are represented, how the scoring function is operationalized, and how the update step is concretely performed across different attack types, are missing. This abstraction limits reproducibility and makes it difficult to assess the actual novelty or technical rigor of the proposed adaptive framework.
3. Some claims are too strong but insufficiently justified. To substantiate such a claim, it would be better to provide a theoretical formulation of robustness in the context of jailbreak attacks, including what constitutes a robust defense, how robustness should be measured under adaptive adversaries, and what characteristics a defense must satisfy to meet that standard.

**Questions:**

1. How are the four steps of the PSSU loop concretely instantiated under each threat model setup? Does the framework support cross-family adaptation (e.g., applying reinforcement learning fine-tuning after a search-based exploration), or are the four attack families evaluated independently?
2. What objective function is used to quantify success, detectability, and cost in the Score step? In the Update step, how are gradient or policy updates performed in the highly discrete prompt space? Are gradients approximated in embedding space or estimated through REINFORCE-like techniques? What are the termination criteria for the optimization loop (e.g., fixed iterations, convergence threshold, or resource budget)?
3. What is the performance of the proposed PSSU loop on the 12 evaluated defenses? Are any of these defenses considered robust under this framework? Additionally, the results presentation appears unclear; if detailed outcomes are shown in Figure 1, this figure is never referenced or discussed in the main text.
4. Evaluation metrics: What metrics are used to measure the success of jailbreak attacks under different threat model setups? The paper mentions using an LLM judge. Does this judge produce a binary success/failure label or a graded score along a numerical range?
5. It would strengthen the paper to include a more comprehensive discussion of existing jailbreak attack methodologies.

---

> ### Author Response · Authors · 2025-11-25
>
> Thank you for the review and questions,
>
>
> Q1: Appendix A, C, D, and E in the submission describe the instantiations and implementation details for each attack type. In brief, for gradient-based attacks we use GCG; for RL-based attacks we use GRPO (details in Appendix C); and for search-based attacks we use a genetic algorithm (details in Appendix D). We would like to reiterate that the primary focus of our work is the overall evaluation setup, not proposing new attack algorithms. For all attack methods we limit the number of rounds, but note that the evaluation considers an attack successful as soon as a single successful attempt is found.
>
> Q2. For RL-based algorithm, the details are in Appendix C as mentioned the algorithm used is GRPO and we used harmbench classifier score and perplexity of secret in datasentinel as score as mentioned Appendix C. Appendix D also has details on the search-based algorithm.
>
> Q3. Figure 1 summarizes the results; we fixed the writing to correctly refer to. Due to space limitations, we only included high-level summaries of each defense in the main text; full results are provided in Section 5 and Appendices C–E. As discussed in the paper, none of the evaluated defenses were robust under our adaptive attacks. If there are specific aspects you found unclear, we are happy to add more detail in the camera-ready version.
>
> Q4.  As stated in the paper, we follow the exact success criteria defined by the original defenses. Unfortunately, there is no coherent evaluation strategy across defenses, and they use a wide range of metrics. However, in all cases the success criterion reduces to a binary success/fail label.
>
> Q5. We will add more detail of the existing attacks to the appendix.

---

### Official Review · Reviewer_8Azb · 2025-10-30

**Soundness:** 4
**Presentation:** 4
**Contribution:** 3
**Rating:** 8
**Confidence:** 4

**Summary:**

This paper critiques current LLM jailbreak and prompt injection defenses by showing that most are evaluated against static or weak attack sets, leading to a false sense of robustness. The authors propose a general adaptive attack framework that integrates gradient-based, reinforcement learning, search-based, and human red-teaming attacks. Using these methods, they systematically break 12 state-of-the-art defenses and achieve over 90% attack success rate in most cases.

**Strengths:**

1. Strong empirical contribution. The authors conduct a comprehensive empirical study, evaluating 12 well-known LLM defenses under multiple adaptive attack paradigms. The scale and depth of this evaluation are impressive and provide much-needed realism in jailbreak defense assessment.

2. Methodological framework. The generalized “PSSU” adaptive attack loop unifies existing attack types under a single conceptual lens. This provides a reusable structure for future evaluation tools.

3. High practical relevance. The results have direct implications for practitioners and researchers designing LLM defenses. The finding that most defenses fail under realistic adaptive evaluation is an important wake-up call for the field.

4. Contribution to jailbreak research. The study serves as a critical meta-evaluation for the entire jailbreak-defense literature and offers clear guidelines (e.g., including human red-teaming, adaptive tuning) that can inspire more rigorous evaluation pipelines.

**Weaknesses:**

1. Cross-method comparability and interpretability. Because the 12 defenses come from heterogeneous benchmarks and threat settings, the reported attack success rates are not directly comparable. The paper explicitly states this limitation, but it weakens the quantitative conclusions — making it hard to decide which defense class performs better. As a result, the framework cannot currently guide defense selection.

2. Limited extensibility and sustainability. The framework lacks an explicit plan for continual integration of new attacks and defenses. If the ecosystem is not maintained or modularized, future defenses or new attack paradigms (e.g., agentic adaptive attacks or multimodal prompts) may again reveal unaddressed vulnerabilities. Without an open, extensible benchmark infrastructure, the community risks repeating the same evaluation pitfalls.

3. Lack of actionable guidance. While the paper convincingly breaks defenses, it provides limited insight into how to design better ones. Beyond “evaluate adaptively,” readers are left without systematic principles for constructing more resilient defenses.

4. Evaluation cost realism. Some adaptive attacks assume substantial compute and fine-tuning budgets. It remains unclear how such evaluations can be standardized or made reproducible across different resource regimes.

**Questions:**

1. After running adaptive evaluations across multiple defenses, how should practitioners select among defenses? Is there a metric or qualitative signal that indicates partial robustness under your framework?

2. Can your adaptive evaluation framework also provide guidance for improving defenses (e.g., by analyzing failure patterns or sensitivity to attack families)?

---

> ### Author Response · Authors · 2025-11-25
>
> Thank you for the review and suggestions. Our primary goal is to highlight that existing robustness evaluations are non-adaptive and insufficient for establishing robustness claims as used in existing defenses. While we introduce an adaptive evaluation framework, we emphasize that no single attack (including ours) should be treated as a definitive robustness metric unless it can be proven to be optimal.
> Instead, our framework as you mentioned in your second question is intended to showcase existing defenses that are vulnerable to several attacks and can help to find common failure modes. However, for future works tailored adaptive attacks should be used to find known failure modes of their defense and we caution against interpreting any finite set of attacks as proof of robustness.
> We believe this perspective is critical for guiding the community toward more principled robustness evaluation practices, even if a complete, universally comparable evaluation framework is not yet achievable.

---

> > ### Comment · Reviewer_8Azb · 2025-11-27
> >
> > Thank you for your response and the additional clarifications. I appreciate the effort; however, the reply does not fully address my original concerns. Considering the contribution and the remaining unresolved issues, I have adjusted my score to a 6.

---

> > > ### Author Response · Authors · 2025-11-28
> > >
> > > Thank you for the response, here is an additional set of experiments to address additional concerns:
> > >
> > >
> > >
> > >
> > > **Comparing multiple defenses**: Our framework can be used to compare defenses within the same threat model and task setting. For defense families that have comparable setups, we already provide direct comparisons (e.g., Table 7). As shown in these cases, all defenses show similarly high vulnerability under adaptive attacks, while numerical differences exist, none are robust in any meaningful sense. Our current takeaway, is that existing defenses uniformly fail under adaptive evaluation, and not that one particular defense class performs reliably better than another. If we had observed any defense providing an order of magnitude greater robustness against our attack, we would have been more comfortable stating that it is better, at least against our specific methods. However, this did not happen in any of our experiments.
> > >
> > >
> > > In fact, we should be caution when  using our results to directly compare the defenses. We don’t believe that defense A with ASR of, says, 58% is meaningfully more robust than defense B with ASR of 76%. The sample size is relatively small (80 for AgentDojo). By modifying our attack algorithm slightly (changing system prompts or attacker LLMs), it is likely that one could discover a different set of successful attacks and a different ASR. Hence, we intentionally avoid drawing a conclusion on the defenses.
> > >
> > >
> > >
> > > **Extensibility and Guidance**: Our general framework can be extended to any defense (specially if we consider common benchmarks such as AgentDojo). However, following our main message, we do not believe that any fixed list of attacks (ours included) should serve as a robustness metric. Unless an attack can be shown to be optimal, the correct practice is for defense designers to construct tailored adaptive attacks that specifically target their mechanism. Our framework provides a structured starting point and helps surface common failure modes, but we emphasize that no finite attack set is sufficient to claim robustness. This is why our guidance for future works is that the defenses should use adaptive attacks.
> > >
> > >
> > > To summarize, our guidance is as follows:
> > > Future defense works should move away from a static set of triggers. Instead, use the strongest available search or optimization algorithm. We believe our search or RL attacks are currently good candidates.
> > > Future defense works should also conduct human red-teaming in addition to the automated attacks.
> > > Future defense works should consider substantially different defense techniques than the ones that we have shown broken and avoid repeating the same mistakes.
> > > Future attack works can look to improve on our automated attacks; we believe that our attacks are far from optimally efficient.
> > >
> > >
> > >
> > > **Cost of evaluation**: Our results show that even without fine-tuning or high-cost optimization, many adaptive attacks already succeed with high rates. Rather than constraining evaluations to low-cost settings, we argue that researchers should report the cost of the attack (compute, queries, or budget), analogous to how traditional security domains report the cost of breaking a system. What can be considered an “acceptable” attack cost varies by application domain for example, a browser agent will have a higher adversary budget than a basic chat interface. Disclosing cost information alongside attack success gives a better understanding of the attacks. We will add further discussion of this perspective in the paper.
> > >
> > >
> > > For the search attack, we estimate the median cost for evaluating one scenario to be about $2 (for the attack algorithm alone, excluding the inference cost of the target system), and it takes about 5 minutes. If we run the attack to finish (100 queries), the cost is ~$50 (and takes 3 hours) per scenario. We have run additional experiments on using cheaper and open-weight models as the mutator and the critic LLMs (see table below). While stronger proprietary models are better attackers, cheaper models (like Gemini 2.5 Flash Lite) and smaller open-weight models (like Qwen 3 32B) are strong alternatives with 1–2 orders of magnitude smaller in cost. As mentioned, we believe there are also rooms for improvement for our attack in terms of efficiency (e.g., reusing triggers from prior runs, different targets, and across scenarios).
> > >
> > >
> > > | Mutator & Critic LLM  | ASR (%) |
> > > |:----------------------|:-------:|
> > > | Gemini 2.5 Pro        | 100     |
> > > | Gemini 2.5 Flash      | 91      |
> > > | Gemini 2.5 Flash Lite | 89      |
> > > | Gemma 3 27B           | 76      |
> > > | Qwen 3 235B A22B      | 100     |
> > > | Qwen 3 32B            | 91      |
> > > | Qwen 3 4B Thinking    | 39      |
> > >
> > >
> > > (ASR is against Gemini 2.5 Pro on AgentDojo)
> > >
> > > For our RL based experiments the cost of evaluating one defense is in the order of a few thousands of dollars but keep in mind we did not perform any optimization to make the attacks more efficient.

---

### Official Review · Reviewer_BBuB · 2025-11-01

**Soundness:** 4
**Presentation:** 4
**Contribution:** 4
**Rating:** 8
**Confidence:** 4

**Summary:**

In this paper, the authors propose a brand-new framework to reliably evaluate the robustness of defense against jailbreak and prompt injection attacks. With the solid evaluation, the authors bypass 12 recent SOTA defenses with ASR higher than 90% for most of the methods.

**Strengths:**

1 This paper is well-written.

2 The soundness of this method is good.

3 The experiments are quite solid.

4 The findings proposed in this paper are interesting to the community.

**Weaknesses:**

From my view, this paper has no obvious weakness, and I think the solid evaluation made by the authors and the insights proposed in this paper should be highlighted to the adversarial community.

**Questions:**

1 How does the proposed framework perform on the Claude-family models, such as Claude 3.5,  Claude 4 and Claude 4.5 Sonnet?


2 In reality, the defenders usually apply multiple defenses instead of a single one. Does combining multiple strong defense methods enhance the empirical robustness of the system?

---

> ### Author Response · Authors · 2025-11-25
>
> Thank you very much for the reviews and suggestions. Here is the answer to questions:
>
>
> Yes we tried on a smaller scale on the Claude family of the model, similar approaches also work on the Claude-family of the models. However, we currently do not have the full agreement with anthropic to run the red teaming at this time yet. But we will add it later to the camera-ready version.
> We agree that in practice multiple of the defenses can be used, but we have seen many of the defenses can be vulnerable to similar vulnerabilities. We evaluated a few combinations of a fine-tuning defense (like MetaSecAlign and most proprietary models) and a detector defense. While the combinations can slightly decrease the ASR, they remain far from secure. We would like to also note that for this work, we argue that existing works proposing defenses should focus on the correct evaluations of their works therefore, we focused on their original setups.

---

> > ### Comment · Reviewer_BBuB · 2025-11-27
> >
> > Thank you for your response, I will maintain my score.

---

### Meta-Review · Area_Chair_AzoM · 2026-01-07

**Summary:**

This paper is on the topic of jailbreaking attacks. This is an area that has seen a major increase of papers in the last 2+ years with many of them reporting high success rate in major LLMs. This paper similarly achieves high success rate over major LLMs and suggests that strong adaptive attacks are sufficient. This paper also contributes that the adaptive attacks can use strong LLMs (but this was already published before) with the idea being that employing gradient-based, reinforcement learning, and search-based attacks the study demonstrates that 12 recent defenses fail to prevent jailbreaks, with success rates exceeding 90%. The paper has extensive empirical comparisons, but as raised by a reviewer, there are some very strong claims that are not backed up by any theoretical results. This is a challenging paper to make a suggestion in the current batch; the results are similar to other recent jailbreaking papers suggesting that they can "trick" the model in the end to reveal harmful information. However, the insights are not tested on the strongest models, i.e. the Claude family, which are typically the hardest to jailbreak, while no new attacks are proposed. Therefore, the lack of any theoretical result (which is not necessary on its own), but also the concerns raised over the setup and the experiments, make it hard to accept the paper.

**Reviewer Concerns:**

The information from the reviews are mixed. For instance, the review from Reviewer BBuB does not offer any relevant information about the paper (summary or positives) and makes one critical question: the vulnerability to the Claude family. The response in the rebuttal is that there was no permission from Anthropic, which is hard to judge on. However, it has been consistently reported that those are some of the strongest models to jailbreak, so it would be very hard to publish a paper with strong claims if the strongest results are not accounted for.

One important concern of Reviewer XpmW is the threat model and the reviewer asked for additional information on those. The rebuttal clarifies that "the primary focus of our work is the overall evaluation setup, not proposing new attack algorithms".

The Reviewer XpmW also raises a point about the consistency of the evaluations and the metrics, which is only partly answered in the rebuttal.

**Reviewer Scores:**

I am not sure if the review scores would have changed. Some of the reviewers have already high scores, but their review was actually not having many positive/negative points, so it would be hard to justify the change of the score.

For the reviewer with the negative score, some of the questions were answered, but it's unclear if they would have been satisfactory. Given those extreme differences, it's hard to judge this paper without a discussion with authors/reviewers and AC/reviewer discussion. Therefore, as AC I do not mind if the decision changes.

---

### Decision · Program_Chairs · 2026-01-26

Reject